# Investigation of Nurses’ Wellbeing towards Errors in Clinical Practice—The Role of Resilience

**DOI:** 10.3390/medicina59101850

**Published:** 2023-10-18

**Authors:** Despoina Pappa, Ioannis Koutelekos, Eleni Evangelou, Evangelos Dousis, Polyxeni Mangoulia, Georgia Gerogianni, Afroditi Zartaloudi, Georgia Toulia, Martha Kelesi, Nikoletta Margari, Eftychia Ferentinou, Areti Stavropoulou, Chrysoula Dafogianni

**Affiliations:** 1Department of Nursing, University of West Attica, 12243 Athens, Greece; 2Department of Nursing, National and Kapodistrian University of Athens, 10679 Athens, Greece

**Keywords:** nurses, errors, wellbeing, resilience

## Abstract

*Background and Objectives*: The fatigue, stress, and burnout of nurses lead to them frequently making mistakes, which have a negative impact not only on the safety of the patients but also on their psychology. The ability to bounce back from mistakes is crucial for nurses. Nursing staff members’ physical and mental health, particularly their depression, is far from ideal, and this ill health is directly correlated with the frequency of self-reported medical errors. The nurses’ mental and physical health are also positively correlated with their perception of wellness support at work. This cross-sectional study aimed to investigate the status of nurses’ mental and physical health regarding clinical errors and the impact of resilience on coping with these situations. *Materials and Methods*: A total of 364 healthcare professionals participated in this research; 87.5% of them were females and 12.5% of them were males. Most of the participants were 22–35 years old. The median number of years of employment was nine. Clinical nurses anonymously and voluntarily completed a special structured questionnaire that included questions from different validated tools in order to assess their state of physical and mental wellbeing after events of stress and errors made during their practice. *Results*: In total, 49.4% of the nurses had made an error on their own, and 73.2% had witnessed an error that someone else had made. At the time of the error, 29.9% of the participants were in charge of more than 20 patients, while 28.9% were responsible for a maximum of three patients. Participants who were 36–45 years old had more resilience (*p* = 0.049) and experienced fewer negative emotions than participants who were 22–35 years old. The participants who mentioned more positive feelings according to their mental state had greater resilience (*p* > 0.001). *Conclusions*: Errors were likely to happen during clinical practice due to nurses’ negative experiences. The level of resilience among the nursing population was found to play a very important role not only in making mistakes but also in coping with errors during their daily routine. Wellness and prevention must be given top priority in all healthcare systems across the country in order to promote nurses’ optimal health and wellbeing, raise the standard of care, and reduce the likelihood of expensive, avoidable medical errors. Healthcare administrations should promote prevention programs for stress occurrence in order to support nurses’ wellbeing maintenance.

## 1. Introduction

The health and wellbeing of clinicians is currently a major priority area in the healthcare context due to the fact that doctors, nurses, and other healthcare providers have a higher risk of compassion fatigue, depression, a poor work–life balance, and suicide than the general population [1,2,3]. The National Academy of Medicine has established an action collaborative on clinician wellbeing and resilience in response to this public health crisis. Despite their reputation for excellent patient care, healthcare professionals frequently neglect their own needs. The quality of treatment provided, and patient safety are both impacted by burnout and practitioners’ less-than-optimal states of wellbeing. Medical errors now account for almost 50% of hospitalized patients and are the third highest cause of mortality in America [4,5].

Several studies indicate that the nursing population faces serious problems with burnout and wellbeing due to workload, office atmosphere, career satisfaction, patient satisfaction, and coping mechanisms [6,7,8,9,10,11]. On the other hand, researchers claim that the individuality of a person is responsible for maintaining a normal emotional status [6,7,8,9,10,11]. When someone refers to wellbeing, he reports the positive and negative feelings about an experience. Positive aspects of wellbeing examination are inspiration, excitement, interest, proudness, strength, and enthusiasm.

On the other hand, the negative feelings are nervousness, guilt, distress, fear, and shame. Good mental health is very important for nurses who are supposed to provide a high quality of patient care [11]. Furthermore, it is essential for enabling them to confront several difficulties within their work environment. Nurses should use recognized techniques to conduct brief self-assessments to gauge their wellbeing, obtain feedback on how they are doing compared to other nurses, and recognize when their discomfort may be negatively affecting their work performance or personal health [12]. 

Nurses’ experiences with errors and the resulting psychological trauma, which frequently increases the risk of providing more subpar care, are receiving more and more attention. Nurses frequently experience negative emotions like fear, guilt, anger, embarrassment, depression, and humiliation [13]. They might experience a loss of confidence, grief, flashbacks, sadness, frustration, and anger [14]. Because there is evidence that psychological resilience components can reduce psychological distress and because there may be low-cost opportunities to help, nurses are becoming more interested in the development and application of resilience-based therapies to address psychological distress [15,16,17,18,19,20]. 

The aim of the study was to assess the physical and mental health status of the nursing population working in clinical departments but also to measure the level of resilience toward making errors. It is very important to investigate the impact of wellbeing on the nursing population because it can contribute to the development of specific strengthening programs and help nursing administrations holistically protect their nurses from negative situations and adverse events. 

## 2. Materials and Methods

### 2.1. Participants

Participants in the study included nurses from all levels of education, including university nurses, nurses from technological institutions, and secondary school assistant nurses. The participants in the current study were general hospital employees in Greece. Specifically, pathological departments, surgical sections, ICUs, respiratory clinics, and oncology departments from four tertiary hospitals were included in questionnaire distribution. 

### 2.2. Data Collection

The research was a cross-sectional study conducted from November 2020 to November 2021 using voluntary, anonymous survey responses. The University of West Attica’s Ethics Committee (52654—20 July 2020) and the scientific councils of all the participating institutions gave their approval to the study. In order to develop a different method of distributing and collecting questionnaires during the entire study period due to the hospitals’ restrictive measures for the global pandemic, an electronic version of the tool was also developed.

### 2.3. Instruments

Four sections made up the research tool: 1. The demographic information: questions about the participants’ gender, age, marital status, educational attainment, and details about their working department (inpatient nurse, outpatient nurse, operating nurse, oncology nurse, or other), as well as the length of time they have worked in a particular unit. 2. The Positive and Negative Affect Schedule (PANAS) [21], which is a positive and negative affect questionnaire that asks participants to rate each emotion’s intensity on a scale of 1 to 5 on a positive and negative affect tool. 3. The Taxonomy of Error, Root Cause Analysis and Practice-responsibility (TERCAP), which was designed to collect nursing practice breakdown data from different American boards of nursing. It outlines a series of classifications based on ideas of excellent nursing practice, including safe medication administration, documentation, surveillance, prevention, intervention, clinical reasoning, interpretation of orders, and professional responsibility/patient advocacy [22]. 4. The Brief Resilience Scale (BRS) [23], which is a 5-point Likert scale about six specific statements of daily life routine. It is essential to mention that the questionnaires were validated in Greek language using double translation, assessing the tool by test re-test performance. Participants were all Greek citizens, so the translation into Greek was the appropriate method for the research tool to be understood. Specifically, there was a special question within the demographics section asking if Greek language was participants’ mother language with 100% positive answers. A written informed consent was signed by all study participants (when natural distribution was possible). In case of electronic completion, participants had the mandatory option to select “Agree’’ or “Not agree” in order to continue or not with the rest of the research tool. 

### 2.4. Data Analysis

The Kolmogorov–Smirnov criterion was used to check the normality of quantitative variables in the beginning. Quantitative variables were expressed as median (interquartile range: IQR) or mean (standard deviation). Absolute and relative frequencies were used to express qualitative variables. While qualitative variables were expressed as absolute and relative frequencies, quantitative variables were expressed as mean values (SD). Student’s t-tests were computed to compare the mean values. The PANAS subscales were used as a dependent variable in multiple linear regression analysis. The participants’ demographics, traits related to their jobs, and the likelihood that an unexpected error occurred while they were performing those jobs were all included in the regression equation. Based on the outcomes of the linear regression analyses, adjusted regression coefficients (β) with standard errors (SE) were calculated. *p* values are always reported with two tails. Analyses were carried out using SPSS statistical software (version 22.0), with statistical significance set at *p* < 0.05.

## 3. Results

The characteristics of the sample, which included 364 Greek healthcare professionals (87.5% of whom were women), are shown in Table 1. Most participants ranged in age from 22 to 35. A total of 50.3% of the participants were married, and 45.6% of them were parents. Furthermore, 10.2% of the participants were specialized nurses, and 47.9% of the participants were university graduates. In their working hospital, the median number of years of employment was nine (IQR: one to fifteen). The average score for resilience was 20.4 (SD: 4.2).

When asked if they had ever made an error while at work, 65.8% (N = 239) of the participants said yes. Nearly half of the participants (49.4%) had made an error on their own, and 73.2% had been the witness of an error that someone else had made. At the time of the error, the median length of employment was two years (IQR: one to five years). The daily shift (39.6%) or the afternoon shift (35.5%) were the times when errors happened most frequently. At the time of the error, 29.9% of the participants were in charge of more than 20 patients, while 28.9% had a maximum of three patients. In 16.3% of cases, the administration had called the participant to inform them of good daily practice, and in 5.3% of cases, this error had negative effects on the participant’s ability to perform their job. 

The mean scores for the subscales measuring positive and negative emotions were 35.2 (SD = 6.44) and 20.08 (SD = 6.82), respectively (Table 2). Additionally, there was no discernible difference in the participants’ positive feelings scores between those who had encountered errors at work and those who had not. Participants who had witnessed an error at work, however, scored significantly higher on the negative feelings scale, i.e., felt more negatively. 

When a multiple linear regression was used, it was discovered that participants with a monthly income of over EUR 1000, those with a second job, and those who were more resilient reported significantly higher levels of positive emotions (Table 3).

After accounting for all demographic and job-related factors (Table 4), participants who had made a mistake at work were found to have more negative feelings than those who had not, and this difference remained substantial. Additionally, it was found that participants who were 36–45 years old and stated to be more resilient experienced fewer negative emotions than participants who were younger (22–35 years old).

## 4. Discussion

The present study was conducted to investigate nurses’ wellbeing following errors in clinical practice. The role of resilience was examined too. In total, 364 nurses in Greece participated. Most of them were women, which is consistent with other studies [24,25,26]. More males are entering the nursing field because of recent developments in society, healthcare, and nursing internationally. The causes of this are unknown, although they can include cultural views on how men and women should behave in society, the standing of nursing, or the wages and working conditions of nurses [23]. The majority of participants ranged in age from 22 to 35. Kahriman et al. [27] reported the average age of nurses to be 33 years, while in Bilgic et al.’s study [25], the average age of nurses was higher (39 years old). 

Nearly half of the participants had made a medical error on their own, and the majority of them had been the victims of a medical error that someone else had made. This is a little lower percentage compared to a study in Ethiopia including 423 nurses, in which more than half of the participants failed in the administration of medication at least once in the 12 months preceding the study [28]. Data from 408 nurses in Saudi Arabia indicated that more than half of them had committed a medication error, but less than half of those nurses were reported, whereas medication mistakes involving incorrect dosages were the most frequent [29]. In the USA, medication errors among nurses are reported to be lower [30]. As a result, variations in rates are brought about by variations in organizational reporting methods and study time periods. 

Although it was not one of the top concerns in our analysis and in the study of Brabcová et al. [31], staff shortages are the second most common cause of prescription administration errors [32]. At the time of the medical error, the median length of employment was two years (IQR: one to five years). According to earlier studies, nurses with little work experience had a higher likelihood of making prescription administration mistakes [29,30,33]. One nursing practice that becomes better with experience is administering medications. Through experience, nurses can develop their abilities and learn more about safe medicine administration techniques.

Additionally, seasoned nurses are familiar with a variety of drugs and techniques [28]. Only in a few cases did the medical error negatively impact the participant’s capacity to execute their job. In almost two out of ten of the situations, the administration informed them of excellent everyday practice. Management should place more emphasis on the system as a potential source of the error than on the individual, as this strategy will raise the standard of care. Staff members are reluctant to disclose mistakes and inappropriate behavior when management takes an authoritative and constrictive approach to them. The top four reasons why medication errors are not reported, in the view of nurses [31], are as follows: fear of accusations, fear of negative reactions from the patient or their family, fear of management reactions, and fear of physician reactions. According to another study [34], medical errors and underreporting are caused by five key causes: individual factors, workplace variables, managerial variables, workplace customs, and mechanisms for reporting errors. It is a moral and legal requirement in every healthcare setting to report medication errors since they are crucial to streamlining the drug management process. Underreporting medication errors is seen as a serious issue since failing to report errors could result in the loss of a valuable source of information. A prior investigation revealed that nurses were underreporting [35]. Over nine out of ten of complaints are self-reported, which adds to the weight of the announcement. Nearly 6000 to 20,000 people died in Taiwan due to drug errors, and one tenth of medical lawsuits were the result of underreporting [36]. In Turkey, almost seven out of ten of nurses who were directly involved in pharmaceutical errors failed to report them [37].

Errors in the administration of medications were discovered to be significantly related to work shifts. Nurses who worked the day or afternoon shift had a higher risk of making drug administration mistakes. According to studies, medical errors mostly happen in the morning [38,39,40] or during the night shift [28,41,42]. Another finding of the study was that greater resilience and fewer negative feelings were reported by adults aged 36 to 45 compared to participants aged 22 to 35. This finding agrees with another study [19], which also demonstrated a beneficial relationship between nurses’ overall health and resilience level in terms of somatic symptoms, anxiety/insomnia, and severe depression. Therefore, nurses who were stronger displayed fewer physical complaints. According to one study’s findings, nurses generally exhibited a moderate level of resilience, with an average score of 63.77 (SD 12.80). Studies conducted in other nations reported findings that were comparable. According to Hegney et al. [43], the mean resilience score for Australian nurses was 70.02, which was slightly lower than the scores for nurse leaders and community samples. Based on Gillespie et al. [44], the development of resilience in nurses has been linked to improved health, a higher quality of life, and efficient use of adaptive coping mechanisms.

Our study revealed normal levels of reported resilience that did not affect the experience of the error. Over 70% of American nurses, according to Mealer et al. [45], exhibit a moderate level of resilience. The relevant earlier studies show that enhancing nurses’ resilience would also increase their job satisfaction, which would lower the global nursing turnover rate [46]. Also, Koen et al. [47] investigated the characteristics of resilience among professional nurses and found that resilient nurses displayed low levels of mental distress. One of the most important psychological aspects relating to employees’ emotional wellbeing and professional performance is resilience [48].

The present study revealed the necessity of the continuation of the research, mostly due to its significant records of nursing errors and the complexity of the factors promoting them. Nurses must be encouraged to complete questionnaires, especially those newly developed, in order to assess their attitude toward mistakes, preventing them, or correcting them, whenever possible.

## 5. Conclusions

Many nurses speak of mistakes that they made while working with patients. However, other nurses record no errors at all during the course of their careers. Errors mostly occurred during the morning and afternoon shifts, as nurses described. The impact of negative emotions on witnessing or making errors was significant in the present study. To prevent or reduce the likelihood of medication errors, nurses must become familiar with a variety of techniques. Hospitals should orient their culture toward nurses who experience mistakes, not only to find a path for better recording of errors but also to develop resilience strategies, assisting the nursing population to provide better quality of healthcare. The core of clinical environments are the nurses, who are urged to work together as an integrated team to reduce the likelihood of errors. Therefore, systematizing the rules is necessary, including education and training, independent checks, standardized procedures, observance of the five rights (medication error prevention), documentation, open lines of communication, informing patients of the procedures they perform, adherence to strict rules, improving labeling and package formats, concentrating on the work environment, reducing workload, avoiding distractions, fixing the flawed system, improving job security for nurses, and fostering a culture of a blame-free workplace.

## 6. Limitations of the Study

This study took place during a period of time when the pandemic had a serious impact on each individual’s daily life. Nurses experienced many negative and unknown situations, and they quite often mentioned states of fatigue and stress. During this condition of constant exhaustion, many of them refused to participate in any kind of research. So, the limited population could mean fewer objective conclusions. Further investigation could probably be beneficial to assess the nursing population’s attitudes toward clinical errors along with their resilience levels.

Furthermore, the distributed questionnaires were self-report tools. Each participant declared his/her answer based on his/her personal thoughts, experiences, and feelings, without necessarily implying that what he/she declared as normal was documented and commonly accepted. Quite often, through self-report tools, situations are underestimated as well as overestimated, with the result that participants’ responses deviate significantly in opposite directions. Of course, the subjectivity of the responses was a significant factor when conducting a survey that was capable of disturbing the level of objectivity of the research data.

Finally, a particularly important limitation was the impossibility of an absolute temporal connection between the error and the reported mental state. Participants were asked to respond and report their experience of a nursing error in relation to their current state of health. They were updated to refer to concurrent conditions whenever possible. However, it was not possible to test this connection in real time. Depending on the perceptual capacity, the corresponding incidents were described.

It is also useful to mention the fact that making a mistake is an uncomfortable and embarrassing situation for a fairly large percentage of people in general. It can cause feelings of guilt and shame, which leads to not reporting and recording them. On many occasions, the nurses experience bullying after such events, and the desire of a participant to declare it in the research is inhibited. It is considered a kind of stigmatization in the workplace, and, especially in our country, a culture of encouraging reporting and a tendency to train to avoid mistakes must be developed.

## Figures and Tables

**Table 1 medicina-59-01850-t001:** Sample characteristics.

	N (%)
Gender	
Men	45 (12.5)
Women	316 (87.5)
Age (years)	
22–35	159 (43.9)
36–45	130 (35.9)
46+	73 (20.2)
Married/Living with partner	182 (50.3)
Children	166 (45.6)
Educational level	
High school graduate	36 (9.9)
2-year college graduate	27 (7.4)
University alumni	174 (47.9)
MSc/PhD holder	126 (34.7)
Specialized nurse	37 (10.2)
Monthly income	
EUR 500–1000	184 (50.5)
EUR 1001–1500	170 (46.7)
EUR 1501–2000	9 (2.5)
EUR 2001 and more	1 (0.3)
Second job	44 (12.1)
Greek native speaker	345 (96.9)
Permanent working condition	225 (62.2)
Years of experience in present hospital, median (IQR)	9 (1–15)
Job Position	
Head Nurse	16 (4.4)
Deputy Head Nurse	23 (6.3)
Nurse	254 (70)
Nurse Assistant	64 (17.6)
Other	6 (1.7)
Number of covered beds in your department of work, median (IQR)	12 (7–20.5)
Number of total beds in your work department, median (IQR)	14.5 (9–30)
Brief Resilience Score, mean (SD)	20.4 (4.2)

**Table 2 medicina-59-01850-t002:** PANAS scales in total sample and by the occurrence of an error in the workspace.

	Total Sample		During Your Professional Career, Has a Medical Error Ever Occurred in Your Working Space?				
			No (N = 124; 34.2%)		Yes (N = 239; 65.8%)		
	Mean	SD	Mean	SD	Mean	SD	*p* Student’s *t*-Test
Positive feelings subscale	35.20	6.44	35.62	6.75	34.99	6.28	0.386
Negative feelings subscale	20.08	6.82	18.81	6.20	20.72	7.04	0.012

**Table 3 medicina-59-01850-t003:** Multiple linear regression results with positive feelings scale as dependent variable.

	Positive Feelings Subscale		
	β ^+^	SE ^++^	*p*
Gender			
Men			
Women	−1.50	1.13	0.186
Age			
22–35			
36–45	0.27	1.01	0.792
46+	1.75	1.61	0.278
Married/Living with partner			
No			
Yes	−1.42	0.97	0.145
Children			
No			
Yes	1.24	1.06	0.245
Educational level			
High school graduate/2-year college graduate			
University alumni	2.66	2.36	0.260
MSc/PhD holder	0.54	2.37	0.821
Specialized			
No			
Yes	0.74	1.36	0.587
Monthly income			
EUR 500–1000			
EUR 1001 and above	1.88	0.86	0.031
Second job			
No			
Yes	2.06	1.05	0.050
Greek native speaker			
No			
Yes	0.48	2.37	0.839
Permanent working condition			
No			
Yes	−1.61	0.94	0.089
Years of experience in present hospital, median (IQR)			
Job Position	−0.04	0.07	0.601
Head Nurse/Deputy Head Nurse			
Nurse	−0.72	1.36	0.594
Nurse Assistant/Other	3.92	2.48	0.115
Number of covered beds in your department of work, median (IQR)	−0.04	0.03	0.110
Number of total beds in your work department, median (IQR)	0.02	0.02	0.276
Brief Resilience Score, mean (SD)	0.40	0.08	<0.001
During your professional career, has a medical error ever occurred in your working space?			
No			
Yes	−0.30	0.77	0.697

+ Regression coefficient; ++ standard error.

**Table 4 medicina-59-01850-t004:** Multiple linear regression results with negative feelings scale as dependent variable.

	Negative Feelings Subscale		
	β ^+^	SE ^++^	*p*
Gender			
Men			
Women	0.37	1.21	0.760
Age			
22–35			
36–45	−2.15	1.09	0.049
46 +	−1.54	1.69	0.363
Married/Living with partner			
No			
Yes	0.18	1.04	0.866
Children			
No			
Yes	1.06	1.14	0.352
Educational level			
High school graduate/2-year college graduate			
University alumni	0.32	2.53	0.899
MSc/PhD holder	1.01	2.53	0.690
Specialized			
No			
Yes	0.02	1.43	0.988
Monthly income			
EUR 500–1000			
EUR 1001 and above	0.08	0.91	0.928
Second job			
No			
Yes	0.30	1.12	0.788
Greek native speaker			
No			
Yes	2.51	2.40	0.297
Permanent working condition			
No			
Yes	1.40	1.01	0.167
Years of experience in present hospital, median (IQR)			
Job Position	−0.03	0.08	0.671
Head Nurse/Deputy Head Nurse			
Nurse	0.98	1.44	0.498
Nurse Assistant/Other	1.53	2.64	0.563
Number of covered beds in your department of work, median (IQR)	0.03	0.03	0.257
Number of total beds in your work department, median (IQR)	−0.02	0.02	0.496
Brief Resilience Score	−0.64	0.09	<0.001
During your professional career, has a medical error ever occurred in your working space?			
No			
Yes	2.05	0.82	0.013

+ Regression coefficient; ++ standard error.

## Data Availability

All the data generated during this study are included in this published article.

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
