# Peer review of "Investigation of Nurses’ Wellbeing towards Errors in Clinical Practice—The Role of Resilience"

_medicina, 2023, doi:10.3390/medicina59101850_

Round 1
Reviewer 1 Report
I read with interest the study presented by Pappa and colleagues on the role of reslience on the nurses' wellbeing and the errors resulted in their clinical practice.
I appreciate the effort in choosing and conducting such a sensitive topic, which possible application beyond the country in which the study was conducted. For me, the topic is of great interest to the readership of the journal, and could possibly extend to the general public readership.
However, the study cannot be published in its current form, and in order to be in a publishable form, authors should address the following:
- The Title of the study is quite confusing, and should be re-written to the reflect its content.
- The Abstract should be written without subsection headings, e.g., background and objectives. Also, the decimal in the abstract should be written with ".", insteatd of ",".
- The Introduction could benefit from expanding on the examined issue. Additionally, to facilitat the readership of the introduction should be divided into paragraphs, each containing a major idea rather than a single-paragraph introduction.
- The Methods require including several details. For example, where did the study take place? how many hospitals were included in the study? What are the inclusion and exclusion criteria?
- It is assumed that the instruments used in the study were in English. Is that the case? If so, how could you ensure that all participants understood the instrument appropriately? If not, please describe the translation and its validation process.
- The presentation of the Results is difficult to follow. Please make sure to refer to each table in the text. The tables should directly follow its mention in the manuscipt, rather than listing all of them together at the end of the results.
- Similarly, the organisation of the Discussion is confusing. The discussion should put the study in context of the current literature, compare and contrast its findings with other studies. Then, the study strengths and limitations, as well as future direction of research should be mentioned Before the conclusion.
- Authors mentioned that consent forms were collected from all participants. Please explain how.
The manuscript would benefit significantly from a professional proofread service.
Author Response
- The Title of the study is quite confusing and should be re-written to the reflect its content.
ANSWER: Ok, Changed to ‘’ The relationship between nurses’ wellbeing, errors, and resili-ence in Greek hospitals’’
- The Abstract should be written without subsection headings, e.g., background and objectives. Also, the decimal in the abstract should be written with ".", instead of ",".
ANSWER: Ok, changed.
- The Introduction could benefit from expanding on the examined issue. Additionally, to facilitate the readership of the introduction should be divided into paragraphs, each containing a major idea rather than a single-paragraph introduction.
ANSWER: Ok. Extra lines to support the ‘’Introduction’’ section added.
The health and wellbeing of clinicians is currently a major priority area in the healthcare context due to the fact that doctors, nurses and other healthcare providers have a higher risk of compassion fatigue, depression, poor work-life balance and suicide than the general population [1-3].
- The Methods require including several details. For example, where did the study take place? how many hospitals were included in the study? What are the inclusion and exclusion criteria?
ANSWER: Details were added to the ‘’Methods’’ section. 4 tertiary hospitals in Greece participated in this study. The inclusion criteria were about the participants to be clinical nurses, not community individuals. The exclusion criteria were about the language spoken by the nurses. Greek and non-Greek nurses who speak and read the Greek language could only participate in the study.
- It is assumed that the instruments used in the study were in English. Is that the case? If so, how could you ensure that all participants understood the instrument appropriately? If not, please describe the translation and its validation process.
ANSWER: The instruments were distributed in Greek language. So, every participant could understand the items. BRS questionnaire was already validated in Greece (Stalikas, A., & Kyriazos, T. A. (2017). The Scale of Positive and Negative Experience (SPANE), Greek Version. Athens: Hellenic Association of Positive Psychology). PANAS questionnaire was already validated in Greece (Daskalou, B., & Sygkollitou, E. (2012). Positive and Negative Affect Scale (PANAS). In A. Stalikas, S. Trivila, & P. Roussi (Eds.), Psychometric Tools in Greece (p. 526). Pedio) and TERCAP questionnaire was validated by our team performing translation/back-translation and test retest procedure with 20 nurses.
- The presentation of the Results is difficult to follow. Please make sure to refer to each table in the text. The tables should directly follow its mention in the manuscipt, rather than listing all of them together at the end of the results.
ANSWER: Ok, modified.
- Similarly, the organisation of the Discussion is confusing. The discussion should put the study in context of the current literature, compare and contrast its findings with other studies. Then, the study strengths and limitations, as well as future direction of research should be mentioned Before the conclusion.
ANSWER: Ok, modified.
The present study was conducted to investigate nurses’ wellbeing towards errors in clinical practice. The role of resilience was examined too.
- Authors mentioned that consent forms were collected from all participants. Please explain how.
ANSWER: A written informed consent was signed by all study participants (when the natural distribution was possible). In case of electronic completion, participants had the mandatory option to select ‘’Agree’’ or ‘’Not agree” in order to continue or not to the rest of the research tool.

Reviewer 2 Report
The subject of the manuscript is very interesting. The approach to manuscript creation is satisfactory. In the methodology, I did not understand that you used a validated survey questionnaire or constructed it yourself. If you compiled the questionnaire together, how was the validation done? The group of respondents is satisfactory in scope and criteria. The obtained results are clearly presented. Appropriate statistical methods and tools well applied. Everything is covered by the discussion. The conclusion is good. The references are in the acceptable range, 15 references are up to five years old.
Author Response
Dear reviewer,
Thank you very much for your very cheering comments. I would like to explain the questionnaire issue. We used PANAS, TERCAP, BRS questionnaires from specific authors, not constucted by us. We just validate the questionnaires in order to use them in Greece. That means that we performed the test re-test at specific number of participants and translation from English to Greek with reverse translation too. We have made also some changes in order to increase the readability of the article complying with the journal guidelines!
Sincerely

Round 2
Reviewer 1 Report
The authors have done a great job addressing my concerns in the paper. They only need to include everything they mentioned in their response to my comment "- It is assumed that the instruments used in the study were in English. Is that the case? If so, how could you ensure that all participants understood the instrument appropriately? If not, please describe the translation and its validation process" to their manuscript.
Well done to the authors on their effort.
Author Response
Thank you very much for your comment. We added some lines to the manuscript (red lines) in order to specify the appropriate procedure.
''It is essential to mention that the questionnaires were validated in Greek language using double translation and assessing the tool by test re-test performance. Participants were all Greek citizens so the translation in Greek was the appropriate method for the research tool to be understood. Specifically, there was a special question within data section if Greek language was participants’ mother language with 100% positive answers. A written informed consent was signed by all study participants (when the natural distribution was possible). In case of electronic completion, participants had the mandatory option to select ‘’Agree’’ or ‘’Not agree” in order to continue or not to the rest of the research tool''.
